# How Servant Leadership Leads to Employees’ Customer-Oriented Behavior in the Service Industry? A Dual-Mechanism Model

**DOI:** 10.3390/ijerph17072296

**Published:** 2020-03-29

**Authors:** Mengru Yuan, Wenjing Cai, Xiaopei Gao, Jingtao Fu

**Affiliations:** 1School of Tourism and Events, Hefei University, Hefei 230061, China; yuanmengru@hfuu.edu.cn; 2School of Public Affairs, University of Science and Technology of China, Hefei 230026, China; gaoxp@ustc.edu.cn; 3Department of Management & Organization, Vrije Universiteit Amsterdam, 1081HV Amsterdam, The Netherlands; 4Management School, Hainan University, Haikou 570228, China; 992771@hainanu.edu.cn

**Keywords:** servant leadership, customer-oriented behavior, organizational identification, vitality, dual-mechanism

## Abstract

Although servant leadership has been acknowledged as an important predictor of employees’ behavioral outcomes in the service industry, there is still no cohesive understanding of the positive association between servant leadership and employees’ customer-oriented behavior (COB). This research, drawing on cognitive affective processing system theory (CAPS), empirically investigates the influence of servant leadership on employees’ COB by exploring two mediators (i.e., organizational identification and vitality). We conducted two studies in China, using a cross-sectional design to survey employees in service-oriented technical organizations (Study 1) and a time-lagged design to survey hospitality employees with frontline service jobs in star-level hotels (Study 2). Across both samples, we found that servant leadership enhanced employees’ COB by simultaneously increasing their organizational identification and vitality. We discuss the implications of these results for future research and practice.

## 1. Introduction

Employees engaging in customer-oriented behavior (COB) are becoming increasingly critical in service-oriented organizations [1]. Practically, increasing numbers of organizations seeking to differentiate themselves on the basis of superior customer service require their frontline service workers to display appropriate behaviors that meet the expectations of customers and deliver the service as well as increase customers’ satisfaction, commitment, and loyalty [2]. In this regard, scholars have investigated the predictors of COB and highlighted the role of certain leadership styles (e.g., transformational leadership and empowering leadership) in fostering the quality of service that employees deliver to their customers. Because a leader is the proximal influencer in social environments, how leader behavior is a model that influences employees’ behaviors is important to understand [3]. Along this line of research, a growing number of studies have suggested the benefits of servant leadership as a useful leadership approach to stimulate employees producing such desirable outcomes as service quality and organizational citizen behaviors [4]. Referring to “developing employees to their fullest potential in the area of task effectiveness, community stewardship, self-motivation, and future leadership capabilities” [5] (p. 162), servant leadership is a people-centered leadership style that prioritizes serving both internal employees and external clients [6]. Leading by servant leadership, frontline service employees go beyond the boundary of self-interest and are held accountable for their customers’ benefits. However, previous studies provide limited evidence of the potential positive association between servant leadership and employees’ COB.

As theories of servant leadership evolve, it is important to specify mediating relationships to better understand the process by which servant leadership translates into employees’ COB. Recent reviews in servant leadership literature highlight that servant leaders can facilitate followers’ desirable outcomes at work through enhancing their positive affect and cognitions [7,8], and further call for empirical examinations on opening the black box of why and how servant leadership exerts influences through these mechanisms due to quite few studies pinpointing this issue [9]. To fill in this research gap, in this study, we tackle the complex intervening mechanism of affect and cognitions between empowering leadership and employees’ COB in the service industry. Specifically, given scholars’ arguments that dual-mechanism explorations rather than single mediator designs can build a more parsimonious and useful picture of the effectiveness of leadership [10], we draw on the cognitive affective processing system theory (CAPS) to explain the processes through which servant leadership leads to COB. CAPS suggests that personal behavior is the result of the confluences of cognitive-emotional factors and external situational factors [11]. That is, all the personal cognitive and emotional elements in the individual system constitute the driving force between external situations and behaviors. Through providing evidence to underline the links between positive leadership styles and employee behaviors via cognitive and affective factors [12], leadership scholars acknowledged that it is more meaningful to investigate the production of behavior from the dual path of cognition and emotion under the theoretical framework of CAPS. Thus, following this line of literature, we draw on the theoretical perspective of CAPS to unfold the influential process of servant leadership on COB.

We first from the affective perspective propose that vitality—an affective component of thriving that refers to individuals’ positive and affective feelings of energy, aliveness, and function [13]—mediates the servant leadership–COB association. There is growing research showing that “the role of leaders or supervisors in promoting thriving at work has been understudied in the extant thriving literature” [14] but the role of servant leadership in enabling employees to thrive has received limited empirical support [15,16]. Servant leadership that provides opportunities for followers’ development [17] enables them to experience a feeling of vitality because of their engagement in their work [18]; therefore, vital employees with a state of emotional arousal are supported to actively engage in desirable behaviors toward their customers (e.g., COB).

We further propose an alternative mediator in terms of cognitive mechanism: organizational identification. Referring to “connection between the definition of an organization and the definition a person applies to him- or herself” [19] (p. 242), organizational identification is considered an intervening mechanism because servant leadership highlights personal connections through developing harmonious relationships with followers. Some evidence has suggested that servant leaders who imbue and reinforce the meaning of serving others enable employees to view the organization as attractive and to strongly identify with the organization from the serving perspective [20]. As a result, employees are more likely to integrate the same value of serving by engaging in serving behaviors (e.g., COB) for the benefit of their organizations.

As shown in Figure 1, two intervening mechanisms (organizational identification and vitality) are proposed to transfer the positive influence of servant leadership on employees’ COB in service-oriented organizations. We conducted two studies to test our hypotheses. In Study 1, we used a sample of employees working in service-oriented technical organizations. In Study 2, we collected data from hospitality employees with frontline service jobs in star-level hotels by employing a time-lagged design, in which the outcome variable (i.e., COB) was assessed by leaders one month after the predicator (i.e., servant leadership) and mediators (i.e., organizational identification and vitality) were assessed.

By focusing on whether and how servant leadership contributes to employees’ COB, we aim to make several contributions to the literature. First, we extend the COB research by incorporating possible influence exerted by servant leaders in stimulating employees’ COB, which responds to scholars’ call for research gauging the impact of servant leadership on diverse employees’ desirable outcomes from a broad perspective. Second, drawing on CAPS, we concurrently examine the two parallel but different mediating pathways. In doing so, we move beyond existing research to provide a comprehensive picture of the servant leadership–COB association. Moreover, we address the call for examining multiple mediators in servant leadership-outcome relations [21]. Finally, in terms of methodology, we conducted two studies to advance the understanding of the benefits of servant leadership in various service-oriented industries.

## 2. Research Background and Theoretical Discussion

Servant leadership is situated as a new field of research approach [22]. The past decade has witnessed growing scholarly attention to the topic of servant leadership, showing it to facilitate individuals’ desirable work- and well-being-related outcomes [9]. Greenleaf [23] initiatively conceptualized servant leadership as containing ten major characteristics: listening, empathy, healing, awareness, persuasion, conceptualization, foresight, stewardship, commitment to the growth of people, and building community. With the development of servant leadership literature, scholars have suggested that the theory of servant leadership responding to resolving the challenges of leadership approaches in the twenty-first century specifically highlights leaders providing service to others and reinforcing recognition of building a better tomorrow for all the employees [24]. We follow this line of research to argue that servant leadership as serving followers by caring and putting subordinates first is consistent with the changing requirements of current and future employee management (e.g., concerning employee development) [25].

In this current research, we are aligned with the conceptualization of servant leadership from Liden et al. [26] including seven main dimensions—i.e., emotional healing, creating value for the community, conceptual skills, empowering, helping subordinates grow and succeed, putting subordinates first, and behaving ethically. Consistent with this conceptualization, we, in the following two independent studies, use the same measures to assess servant leadership in the service organizations (see the measurement section below). As servant leadership highlights the service-oriented activities in the workplace [3], scholars have continuously suggested that servant leadership plays a critically important role in managerial activities in the service industry [3,27]. Specifically, servant leadership theory postulates that servant leaders prioritize people more than production [28], which can help service employees relieve their stress and negative emotion during working hours [3]. In this regard, servant leadership is different from other HR actives and leadership approaches (e.g., empowering leadership). Therefore, we, in the current research, aim to extend this stream of servant leadership investigations by testing its effect on employee COB.

We primarily draw on the CAPS to further explore the mechanisms through which servant leadership can contribute to employees’ COB. Theoretically, CAPS argues that situational factors can stimulate both individuals’ cognitive and affective units, which in turn indicate corresponding behaviors [11,29]. The basic principles of CAPS have received increasing support in a wide range of leadership studies [12]. Leadership researchers have found that leadership styles can act as a prominent situational indicator to dynamically activate employees’ cognition and affect system concurrently [30], because employees usually use their cognition and affect to process information from their leaders. Although some studies have found evidence that servant leadership can boost followers’ motivational elements [25], affect and cognition mechanisms receive less empirical attention [9], and considering both the mechanisms simultaneously is relatively rare. In order to extend the well-examined mechanisms (e.g., motivation) in explaining the servant leadership influences, this current research adopts both the cognitive and affective system from the CAPS perspective. Specifically, in two independent studies, we propose and examine whether and how servant leadership can facilitate employees’ COB through promoting their cognitive (i.e., organizational identification) and affective characteristics (i.e., vitality).

## 3. Theory and Hypotheses

### 3.1. Servant Leadership and Employees’ Customer-Oriented Behavior

Servant leadership is initially and significantly characterized as having a sense of service [3]. That is, a servant leader strongly believes that serving others (e.g., subordinates and customers) is his or her primary task, and then, the service consciousness naturally stimulates his/her leadership behavior [31]. Specifically, servant leadership has three distinct characteristics: being oriented toward others, individually considering the unique needs of employees, and focusing on the organization rather than focusing on themselves [9]. At the workplace, servant leaders put their employees first and understand the psychological and material needs of each employee to create an environment in which these employees can realize their potential. In doing so, employees are more likely to perform well at work while achieving their own growth [32]. 

According to previous research that suggests the benefits of servant leadership in facilitating employees’ desirable outcomes in the service industry [3], we propose a positive relation between servant leadership and employees’ COB. Theoretically, COB refers to the goal of customer satisfaction when providing services [33,34], prioritizing customer interests [35] based on customer needs, and the timely adjustment of service behavior [36]. Research has indicated three major factors that affect COB: the surface characteristics of the service personnel, the service atmosphere, and the stability of the service staff’s emotions. Thus, servant leadership, as a contextual factor with an orientation toward providing service, is expected to foster employees’ intention to provide service to their customers.

Empirical studies show that employees who feel emotional identity are willing to put more efforts into completing work (e.g., serving their customers), showing stronger COB [37]. Conversely, if employees are not treated and supported reasonably, they are likely to generate negative service behaviors [1]. Given that the service behavior of service personnel toward customers is directly influenced by the service atmosphere, a positive service atmosphere enables service personnel to provide better services [38]. Furthermore, employees normally treat their leader as a role model and thus follow the behaviors of the leader. Therefore, employees who are managed by a servant leader are more likely to ignore their own interests and prioritize those of others (e.g., customers) [39]. That is, employees with this service orientation tend to provide high-quality service to satisfy their customers. Thus, we hypothesize the following:

**Hypothesis** **1.**
*Servant leadership is positively related to employees’ customer-oriented behavior.*


### 3.2. Vitality as a Mediator

Employee vitality has received considerable attention in researching the influence of leadership styles and employee outcomes, especially the impact of servant leadership [15,40]. It can be generally defined as an affective component of thriving that refers to individuals’ positive and affective feelings of energy, aliveness, and function [13]. As an important aspect of an individual’s emotion and affect, vitality is a kind emotional response experienced by employees when interacting with important surroundings in the workplace. Theoretically, vitality is manifested by psychological strength, emotion, and recognition. These three aspects indicate that individuals with high levels of vitality have a good physiological state and empathize with the needs of others and the need to provide emotional support for others [41]. Previous studies have indicated that when individuals have a high level of vitality, they obtain the “experience of having energy available to one’s self” [42] (p. 356), which is associated with a range of positive health and wellness indicators (e.g., work engagement and overall well-being) [43]. Given that vitality represents an independent and positive emotional response to an employee’s interaction with specific elements of the work environment [44], previous studies have shown that such work contexts as leadership and supervisory behaviors can help employees experience vitality at work [18].

Following existing research, we propose that servant leadership can promote employees’ sense of vitality in the service work environment. Specifically, as servant leaders emphasize providing opportunities to facilitate employees’ personal growth [9,23], employees receive strong socio-emotional support from their servant leader. Thus, employees obtain more attention [15] to overcome obstacles and recover from any energy loss. In this situation, these employees tend to feel vitality at the workplace. Moreover, when subordinates are supervised by their servant leader, they are more likely to experience vitality because they are fully supported with opportunities to engage in their work. According to the above-mentioned reasoning, we can expect that servant leadership can significantly foster employees’ vitality. Thus, we hypothesize the following:

**Hypothesis** **2.**
*Servant leadership is positively related to employees’ vitality.*


Emerging research suggests that vitality at work relates to a number of positive outcomes among employees, such as better task performance, low burnout or stress, job satisfaction, and organizational commitment across different industries [45,46]. More specifically, scholars with empirical evidence have highlighted the importance of vitality among service employees [47] by showing that vitality in the service industry refers to the positive feeling marked by the subjective experience of having energy [48]. As service personnel suffering from excessive workload and job demands are highly prone to stress at work, their positive affect has an important role in helping them relieve the pressure. In the current research, we propose a positive impact of employee vitality on COB.

Previous studies with accumulated relevant evidence support our expectations of the positive association between vitality and COB among service employees. Specifically, Fisher [49] showed that retail clerks’ positive moods can significantly stimulate them to engage in helping behaviors toward coworkers and customers. Moreover, George [50] conducted a study on salespeople of a large retail store and found that employees with higher positive moods were more likely to provide assistance to their coworkers and customers. Following this line of research, an enhancement in vitality levels may increase service employees’ behavioral attempts to complete their tasks. Specifically, since the main task of service employees is to provide customers with superior service, employees with a stronger sense of vitality are more likely to address the difficulties in the workplace to better serve customers. Accordingly, we expect that vitality can facilitate employees’ COB. Thus, we hypothesize the following:

**Hypothesis** **3.**
*Vitality is positively related to employees’ customer-oriented behavior.*


Summarizing the previous theorizing, servant leadership has a positive impact on employees’ sense of vitality by improving their work engagement. As a result, these employees are more likely to perform better during work by displaying a high level of COB. Therefore, we propose a mediating effect: servant leadership contributes to employees’ COB by enhancing their vitality. Therefore, we hypothesize the following:

**Hypothesis** **4.**
*Vitality mediates the relationship between servant leadership and employees’ customer-oriented behavior.*


### 3.3. Organizational Identification as a Mediator

Conceptually, scholars have suggested the potential influences of leadership styles and/or supervisory behaviors on employees’ self-concepts within the organization [51], such as personal identification with their organizations. Specifically, since employees are involved in relationships with their leaders [52], the behaviors and attitudes of their leaders in the organization act as contextual cues, which not only enables employees to recognize their work environments but also changes their orientation from self-interest to the collective interest of the organization [53,54]. In this regard, employees tend to feel that they are valued by their supervisors who represent their organizations; therefore, they (i.e., employees) develop a strong identification with their organizations.

Consistent with the line of research that suggests that there is a relationship between desirable leadership styles and employees’ organizational identification, we propose the positive influences of servant leadership on employees’ organizational identification. Specifically, servant leaders depict the organization as building and promoting an environment to facilitate followers’ personal development and their positive perception of their organization’s image. As a result, they (i.e., employees) are more likely to view the organization as attractive and to strongly identify with the organization [8]. Moreover, as leaders personify the organization [55], servant leadership helps reinforce the connection between employees and the organization [56]. When a leader displays a servant-oriented leadership approach, his/her behaviors (e.g., providing concerns, supports, and resources) make employees feel that their personal needs are satisfied. Consequently, employees develop a strong organizational identification. Thus, we hypothesize the following:

**Hypothesis** **5.**
*Servant leadership is positively related to employees’ organizational identification.*


Theory and research have argued that there are benefits of organizational identification on interpersonal behaviors such as organizational citizenship behaviors and cooperative behaviors [57]. Organizational identification can increase opportunities to strengthen employees’ positive attitudes derived from their role at work [8]. That is, when an employee strongly identifies with his/her organization, he/she is more likely to feel the meaningfulness of the work, which in turn stimulates him/her to exert more energy and effort at work [58]. Specifically, as service-oriented tasks require a high level of employee involvement [1], employees tend to display COB. In addition, as organizational identification signifies the attachment between employees and organizations, scholars have found that it significantly develops individuals’ positive emotional attachment to an organization [59]. According to previous studies, consistent positive emotions are associated with one’s ability or motivation to serve customers well.

Moreover, leaders not only represent the managerial philosophy of their organizations but also act as role models to influence their followers’ behaviors and attitudes. Servant leadership contains service-oriented behavior signals that the organization also favors serving others [6]. In this situation, employees with a high level of identification with their organizations treat their servant leaders as role models by observing, imitating, and following their (i.e., leaders’) behaviors and mindset; therefore, these employees, in the workplace, tend to engage in servant-oriented behaviors by providing more and better service to satisfy customers. Thus, we hypothesize the following:

**Hypothesis** **6.**
*Organizational identification is positively related to employees’ customer-oriented behavior.*


The previous literature has suggested that the association between leadership or supervisory behaviors and employees’ behavioral outcomes are mediated by employees’ self-concept (e.g., rational identification) [53]. Based on the previous arguments, we further propose a mediation effect. That is, employees’ organizational identification acts as a mediator to link the relationship between servant leadership and their COBs. Specifically, servant leadership exerts a positive effect on fostering followers’ organizational identification, which in turn stimulates them to deliver higher quality service to customers. Thus, we hypothesize the following:

**Hypothesis** **7.**
*Organizational identification mediates the relationship between servant leadership and employees’ customer-oriented behavior.*


## 4. Study 1

### 4.1. Sample and Procedure

Through using a questionnaire survey designed for this study, data collection was conducted in three Chinese firms which provide technological service. We first contacted the CEOs/HR officers of these companies to obtain permission for our investigation. Next, employees received a brief introduction explaining that all information provided would be kept confidential, and the results would be sent to the researchers only. To avoid response bias, the names of the measures were not revealed, and the survey was anonymous. Informed consent was obtained from all participants to ensure that the researchers had the right to use the collected data. We asked employees to rate servant leadership, vitality, and their own COB through an online survey. The sample included 288 participants (see Table 1), 54.5% of whom were male (standard deviation (SD) = 0.50). The average age of the employees was 30.5 years (SD = 5.66), and the majority had at least a bachelor’s degree (55.9%) (SD = 0.74). Participants’ average number of years working in the relevant company was 3.3 years (SD = 3.15).

### 4.2. Measures 

We used the translation–back translation procedure to translate all the scales from English to Chinese [60]. Respondents rated each of the items on a 5-point scale from 1 = strongly disagree to 5 = strongly agree (See Table A1. In the Appendix A). 

Servant leadership (Independent Variable): Employees assessed their managers’ servant leadership on a 7-item scale (χ^2^ [11] = 16.98; TLI = 0.99; CFI = 0.99; RMSEA = 0.04) developed by Liden et al. [26]. A sample item is “my manager puts my best interests ahead of his/her own” (Cronbach’s α = 0.87).

Vitality (Mediator): Vitality was evaluated by employees with five items (χ^2^ [3] = 12.78; TLI = 0.95; CFI = 0.99; RMSEA = 0.05) from Kark and Carmeli [13]. A sample item is “I am most vital when I am at work” (Cronbach’s α = 0.86).

COB (Dependent Variable): We used the six-item scale (χ^2^ [7] = 19.25; TLI = 0.97; CFI = 0.99; RMSEA = 0.06) from Peccei and Rosenthal [61] to rate employees’ COB at the workplace. A sample item is “I am always working to improve the service I give to customers” (Cronbach’s α = 0.88). 

Control variables: Consistent with previous studies [25], we controlled for employees’ gender, age, education level, and work tenure.

### 4.3. Results

As all the data were collected from one source at a single point in time, we followed the explanatory factor analysis [62] to identify the potential for common method bias (CMB). The results showed that one factor accounted for 36.64%, which is below the accepted threshold of 40%.

We conducted a confirmatory factor analysis (CFA) to examine the distinctiveness of the key variables in the current study. The results showed adequate fit: χ2 [111] = 262.31, *p* < 0.01; CFI = 0.95; IFI = 0.95; RMSEA = 0.07, which demonstrated a better fit to the data than alternative models: two-factor model (combine servant leadership and vitality) (χ2 [117] = 438.09; CFI = 0.88; IFI = 0.88; RMSEA = 0.10), and one-factor model (combine all three factor) (χ2 [118] = 833.63; CFI = 0.74; IFI = 0.74; RMSEA = 0.15). These results provide support for the distinctiveness of the four study variables for subsequent analyses [63].

Table 2 shows the descriptive statistics, reliabilities, and correlations of all the variables in the current study.

The KMO and Bartlett’s test was applied to check the sample size. The results show that the value of KMO is 0.88, which is greater than the acceptance of 0.77 [64]. Therefore, the sample size is adequate for factor analysis. We then conducted analyses on internal reliability and convergent validity measures in Study 1 (see Table 3).

Table 4 shows the results of testing the mediating effect of servant leadership on employees’ COB through vitality. Specifically, in Model 1, servant leadership (i.e., independent variable) was significantly related to employees’ COB (i.e., dependent variable) (*β* = 0.22, *p* < 0.001, △*R*^2^ = 0.07), thus supporting H1. In Model 2, servant leadership was significantly related to employees’ vitality (i.e., mediator) (*β* = 0.39, *p* < 0.001, △*R*^2^ = 0.20), thus supporting H2. In Model 3, vitality was found to be positively related to COB (*β* = 0.45, *p* < 0.001, △*R*^2^ = 0.21), lending to support H3. Moreover, in Model 4, when servant leadership and vitality were both entered, vitality was positively related to COB (*β* = 0.43, *p* < 0.001, △*R*^2^ = 0.15) while servant leadership was not significantly related to COB (*β* = 0.06, *n.s.*). Therefore, vitality was tested to fully mediate the relation between servant leadership and COB. To further clarify the mediation effect, we used a bootstrap procedure with 10,000 samples to produce a confidence interval (CI) for the indirect effect. The results reveal that the indirect effect through vitality was significant (indirect effect = 0.18, *p* < 0.01, 95% CI (0.12, 0.26)). Therefore, H4 was fully supported.

### 4.4. Discussion

In sum, the results of Study 1 supported some of our hypotheses (H1, H2, H3, and H4), thus, another study is required to replicate and extend the findings in Study 1. Regarding the methodology limitation, Study 1 is a cross-sectional study which leads to a problem of hypothesized causality; therefore, collecting data at different points in time would increase the rigorousness of the design. Moreover, most of our data, including the COB measure, are self-reported from employees, leading to the possibility of bias of self-assessment stemming from personality bias. Although the results of the CMB tests suggested that we should not be concerned with this issue, we considered that it is useful to gather objective data on COB (e.g., supervisor-rating) in order to further establish our results. Third, although we initially proposed to examine the effect of servant leadership on COB in the service industry, our sample in Study 1 was very specific to the organizations which provide technological service, which limits the validity and generalizability of our results. Finally, since CAPS theoretically highlights the potential mechanisms in terms of personal cognition, extra studies are needed to consider the mediating role of cognition-related factors (e.g., vitality). Therefore, using a sample drawn from distinct organizations which provide different services, such as hotels, could increase validity and generalizability.

## 5. Study 2

### 5.1. Sample and Procedure

Data were collected from employees and their direct supervisors working in four- and five-star hotels in China using anonymously completed questionnaires at two different times. Before submitting our questionnaires, we contacted the HR departments of these hotels to express our research topics and aims. After receiving their confirmations, we asked them to provide a name list of employees and their (i.e., employees’) direct supervisors. Afterward, one of the authors submitted the questionnaires to the employees and their supervisors. The surveys were first submitted to the 270 frontline employees to measure servant leadership, vitality, organizational identification, and their personal information during work hours. We received 212 usable surveys back, giving us an 84.8% response rate. One month after the initial survey, we distributed a separate rating form to each of the 35 relevant supervisors asking them to evaluate their subordinates’ COB and their personal information. We received 31 usable responses corresponding to 182 employees (see Table 5), achieving an 88.6% response rate.

The average age of the supervisors was 30.71 years old (SD = 3.84), and 64.5% were male. The reported average tenure in a management role was 4.16 years (SD = 2.66). Most of them (58.1%) hold a bachelor’s degree (SD = 0.65). Among the frontline workers, the average age was 25.17 years old (SD = 4.10), and 58.8% were female. The reported average work tenure was 3.31 years (SD = 3.83). Most of them (47.8%) graduated from high school or a technical secondary school. 

### 5.2. Measures

Unlike Study 1 using the cross-sectional research design, Study 2 used the time-lagged research design. A temporal separation technique was utilized to separate the independent variable from the dependent and mediator variables. In addition, to avoid the CMB, we asked the supervisors to rate their followers’ COB. 

Servant leadership (Independent Variable): We used the same scale [26] as in Study 1 (χ^2^ [12] = 22.46; TLI = 0.97; CFI = 0.97; RMSEA = 0.04) (Cronbach’s a = 0.86).

Vitality (Mediator): We used the same scale [13] as in Study 1 (χ^2^ [3] = 10.11; TLI = 0.95; CFI = 0.99; RMSEA = 0.05) (Cronbach’s a = 0.87).

Organizational identification (Mediator): We used the five-item scale (χ^2^ [5] = 14.69; TLI = 0.98; CFI = 0.98; RMSEA = 0.05) from Smidts and Pruyn [65]. A sample item is “I feel strong ties with my organization” (Cronbach’s α = 0.89).

COB (Dependent Variable): We used the same scale, Peccei and Rosenthal [61], as in Study 1 (χ^2^ [8] = 18.70; TLI = 0.98; CFI = 0.98; RMSEA = 0.04) (Cronbach’s α = 0.86). 

Besides the control variables from employees in Study 1, we also controlled for supervisors’ age, gender, education level, and tenure in a management role in Study 2.

Given that the data were hierarchical, with employees nested in groups, we employed the hierarchical linear modeling (HLM) analyses to test hypotheses. The viability of the construct created through aggregation—servant leadership (aggregated across multiple employees of the same team)—was assessed. We assessed inter-rater agreement by calculating *r*_wg_ [66], and intra-class correlation (ICC) [67]. ICC(1) and ICC(2) values of servant leadership were 0.52 and 0.87 respectively (*p* < 0.001), and the mean *r*_wg_ values of servant leadership were all above 0.95. The results indicate that aggregation is justified.

### 5.3. Results

Before hypotheses testing, we conducted a CFA to examine the distinctiveness of the key variables in the current study. The results showed adequate fit: χ2 [139] = 508.18, *p* < 0.01; CFI = 0.90; IFI = 0.90; RMSEA = 0.08), which demonstrated a better fit to the data than alternative models: three-factor model (combine vitality and organizational identification): (χ2 [249] = 730.90, *p* < 0.01; CFI = 0.81; IFI = 0.82; RMSEA = 0.01), two-factor model (combine servant leadership, COB, and vitality, organizational identification) (χ2 [251] = 827.1, *p* < 0.01; CFI = 0.78; IFI = 0.78; RMSEA = 0.11), and one-factor model (combine all three factors) (χ2 [252] = 843.64, *p* < 0.01; CFI = 0.78; IFI = 0.78; RMSEA = 0.11). These results provide support for the distinctiveness of the four study variables for subsequent analyses [63].

Table 6 represents the descriptive statistics, reliabilities, and correlations of all the variables in the current study.

The KMO and Bartlett’s test was applied to check the sample size. The result show that the value of KMO is 0.90, which is greater than the acceptance of 0.77 [64]. Therefore, the sample size is adequate for factor analysis. We then conducted analyses on internal reliability and convergent validity measures in Study 2 (see Table 7).

Table 8 shows the results of the multilevel influences of servant leadership on COB via vitality and organizational identification. Before testing the hypotheses, we ran a null model to examine the significance of systematic between-group variance. The results show that the chi-square test is significant (χ^2^ [29] = 287.69, *p* < 0.001), supporting the use of HLM. In Table 8, Model 3 shows that servant leadership (i.e., independent variable) is significantly related to employees’ COB (γ = 0.53, *p* ≤ 0.001), thus supporting H1. Moreover, servant leadership is positively related to both vitality (i.e., mediator) (Model 1) (γ = 0.29, *p* ≤ 0.05), and organizational identification (i.e., mediator) (Model 2) (γ = 0.29, *p* ≤ 0.05); therefore, H2 and H5 are both supported. In Model 4, after controlling the variables at the team level, both vitality and organizational identification are positively related to COB (i.e., dependent variable) (γ = 0.58 and γ = 0.23 respectively, *p* ≤ 0.001), supporting H3 and H6. 

In Model 5, both servant leadership (γ = 0.21, *p* ≤ 0.01) and vitality (γ = 0.56, *p* ≤ 0.001) are significantly related to COB, lending support for H4. Meanwhile, both servant leadership (γ = 0.21, *p* ≤ 0.01) and organizational identification (γ = 0.29, *p* ≤ 0.05) are significantly related to COB, lending support for H7. Bootstrapped CIs corroborate the significant indirect effects of servant leadership on COB through vitality (CI_95%_ = (0.12, 0.33)) and through organizational identification (CI_95%_ = (0.05, 0.19)), again supporting both H4 and H7.

### 5.4. Discussion 

Study 2 aimed to replicate and extend the findings of Study 1. The results reveal that servant leadership positively relates to COB by simultaneously influencing employees’ vitality and organizational identification. Rather than the full mediation effects of vitality in Study 1, Study 2 found that both vitality and organizational identification partially mediate the association between servant leadership and COB. Study 2 improved on Study 1 both methodologically and statistically, and the similarity of the two sets of results increased our confidence in their generalizability and validity. Table 9 shows the conclusion of testing all the hypotheses in both Study 1 and Study 2.

## 6. Discussion

### 6.1. Theoretical Implications

Our research contributes to the current literature in several important ways. First, we are among the first to investigate the servant leadership–COB association. We found that servant leaders can generate a significant impact to boost employees’ COB. Moving beyond the previous research suggesting the potential benefits of servant leadership in service industries [3], we enrich the understanding that the people-centered nature of service-oriented organizations requires leaders and managers to engage in servant leadership, which in turn facilitates employees to provide customer-oriented service. As such, our results not only reinforce the need to consider the importance of servant leadership to understand employees’ behaviors in providing high-quality service to their customers, but also empirically respond to scholars’ calls for research by gauging the impact of servant leadership on diverse employees’ desirable outcomes from a broad perspective.

A second and important contribution of our research is enriching our knowledge by unravelling the underlying black box of how servant leadership effectively contributes to employee COB; that is, through applying CAPS, we identified a dual-mechanism process by which servant leadership contributes to employees’ COB. Specifically, servant leadership is a significant stimulus that enables followers not only to identify more with their organization but also to feel more vitality, which both in turn arouse employees’ engagement in COB. That is, we provide multiple lenses to comprehend the influence of servant leadership. Generally, we respond to scholars’ recent calling that “by examining multiple…mediators concurrently, we can rule out some of these effects and build a more parsimonious and useful picture of what is going on.” [10] (p. 558) through utilizing CAPS to test two influential paths of servant leadership on employee COB. Specifically, as CAPS theoretically indicates that situational factors can trigger individuals’ cognitive and affective characteristics towards certain behavioral outcomes [11], leadership scholars have suggested that positive leadership styles are beneficial situational factors to boost employee desirable outcomes at work through influencing employees’ cognitions and affects [12]. The current research extends this line of literature to indicate that servant leadership is characterized as a “positive” leadership approach and can contribute to employees’ positive behaviors (i.e., COB) in the service setting. 

Moreover, through identifying two distinguished mediators simultaneously from both cognitive and affective perspectives, our findings enrich the knowledge of applying CAPS in the servant leadership literature. Regarding the mediator of organizational identification, our results contribute to a social identity approach to (servant) leadership, which has tended to focus on organizational identification [68]. Specifically, by demonstrating the incremental development of employees’ identification with their organizations as well as COB, this research provides insights into the role of identification-related cognitions in the servant leadership process. Regarding the mediator of vitality, our results highlight the importance of emotional characteristics in linking the association between servant leadership and subsequent outcomes [9]. That is, our findings imply that servant leadership is a proximate antecedent of followers’ experience of vitality. Therefore, responding to the conceptual argument that exploring multiple mediators can provide a more parsimonious and useful picture of the effectiveness of leadership [10], we address the call for examining multiple mediators in servant leadership-outcome relations [21]. In this regard, this research deepens our understanding empirically regarding two variables jointly playing a complete mediating role in the theoretical model of servant leadership and COB.

Finally, in terms of methodology, we conducted two studies to advance the understanding of the benefits of servant leadership in various service-oriented industries. Although previous studies have shown the benefits of servant leadership and the significance of COB in the service industry, most of the findings are for the hospitality industry, which limits their generalizability. In the current research, we conducted two studies with different samples from different service-oriented organizations to further support the result that servant leadership fosters employees’ COB. The results consistently indicate that the acknowledged requirement of servant leadership and COB should be highlighted in organizations in which employees’ main task is providing customers with high-quality service and enhancing customers’ satisfaction. Furthermore, as the service workload of these employees is heavy, servant leadership principles are highly valued as operational philosophies for various service-oriented companies.

### 6.2. Practical Implications

According to the findings, the research provides some suggestions for the service-oriented organizations. First, the significance of the servant leadership approach should be highlighted in a wide range of the service-centered organizations. That is, organizations are encouraged to provide training and mentoring programs for managers, which aim to enhance their abilities of enacting servant leadership style. For example, organizations should adopt a servant philosophy and establish servant requirements to develop leaders with a key “servant” orientation and mindset (e.g., emphasizing concerns for followers). Meanwhile, a 360-degree leadership assessment can be launched to evaluate leaders’ or managers’ servant-oriented behaviors and attitudes. In addition, such managerial activities as employees’ motivation surveys, with feedback to management, and embedding results in yearly objectives can be installed in the service-centered organizations to further strengthen supervisors enacting the servant leadership approach.

Next, given the crucial mediating role of vitality, organizations should also increase employees’ vitality. For example, more job resources should be provided to facilitate employees with more energy. Moreover, to promote employee COB, organizations should foster individual, interpersonal, and organizational identities. Beyond initiating organizational-level employee development policies, organizational leaders should assure that employees are aware of organizational investments so that they attribute organizational development practices rightfully to the organization. Therefore, the resulting feelings of obligation should generate COB. Finally, given the both significant mediation effects of vitality and organizations’ identification on the servant leadership–COB association, organizations should emphasize on combining the practices of facilitating the above two together to fully realize the effective functioning of servant leadership.

### 6.3. Limitations

There are some limitations. First, although we used a more rigorous research design in Study 2 to replicate the results in Study 1, we collected data of the independent and mediation variables at the same time (Time 1). The inference of causation among the variables thus should be explained with caution. Future research is encouraged to collect data longitudinally to understand the dynamic of the relationships among these variables. Second, both of our studies took place in China and it is not yet clear whether our results are generalizable to other countries. Future research is needed to test the model in other countries, and it may be particularly interesting to compare the results among countries. Relatedly, although our sample in both studies reflects the various organizations in service environments, using samples from other companies (e.g., airline company) could broaden the implications in the whole service industry. Third, given that Study 1 illustrates a full mediation effect but Study 2 a partial mediation, future research should explore the explanations to justify the servant leadership–COB relationship (e.g., testing the sample differences between different service-oriented organizations). Relatedly, regarding the mediating effect of vitality, although the current study, consistent with previous research, indicated that leadership styles can influence subordinates’ vitality [69], scholars have found that bad management are likely to spoil vitality. For example, Pick and coauthors [70] evidenced that leaders and human resources departments providing training and development are unlikely to foster employees’ vitality because these employees may experience training as too demanding or too time consuming. Therefore, future empirical studies are highly encouraged to investigate the potential dark side of leadership on employee vitality. 

In addition, although the current research including two independent studies empirically examined the dual-mechanisms in the servant leadership–COB relationship, we did not consider other potential mechanisms. Specifically, as scholars in the servant leadership have found the motivational mechanism [25], future studies are highly encouraged to take this mechanism into consideration to further enrich the influencing process of servant leadership. Finally, an objective measure of employees’ COB was not used. In the current research, we relied on subjective assessment of COB (employee self-rated in Study 1 and supervisor-rated in Study 2), which failed to entirely preclude the possibility of personal bias. Given that COB highlights the service customers received from service employees, future studies could involve them (customers) to rate employees’ COB.

## 7. Conclusion 

Researchers have indicated that servant leadership is an essential predictor of employees’ behavioral outcomes in the service industry. However, little effort has been made to examine how servant leadership can facilitate employees’ COB. The current research including two independent studies aims to address this limitation. Specifically, through drawing on CAPS, the research empirically explores two mediators (i.e., organizational identification and vitality). Across both samples in various service-oriented organizations, the findings show that servant leadership enhances employees’ COB by simultaneously increasing their organizational identification and vitality.

## Figures and Tables

**Figure 1 ijerph-17-02296-f001:**
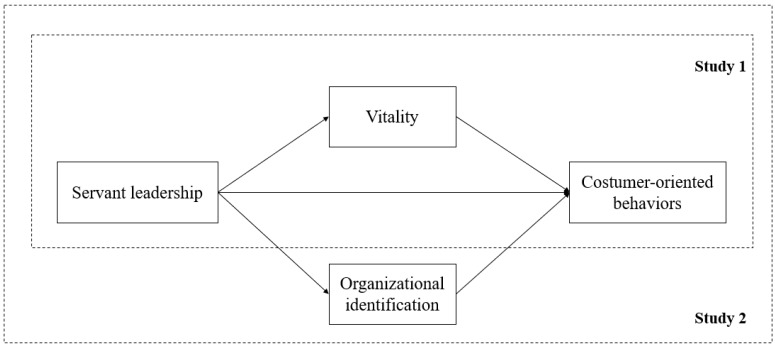
The hypothesized model.

**Table 1 ijerph-17-02296-t001:** Sample distribution (Study 1).

Variables	Value	Frequency	Percent
Gender	Male	157	54.5%
	Female	131	45.5%
Education	High school/technical school	8	2.8%
	Associates degree	83	28.8%
	Bachelors degree	161	55.9%
	Masters degree and above	36	12.5%
Work tenure	<5	208	72.2%
	5–9	70	24.3%
	10–14	5	1.7%
	>15	2	1.7%

**Table 2 ijerph-17-02296-t002:** Descriptive statistics, reliabilities and correlations of variables (Study 1).

Variables	Mean	SD	1	2	3	4	5	6	7
1. Gender	1.46	0.50							
2. Age	30.14	5.66	−0.13 ^*^						
3. Education	3.79	0.74	0.12 ^**^	0.07					
4. Work tenure	3.36	3.15	0.04	0.69 ^**^	−0.05				
5. Servant leadership	3.61	0.66	0.03	0.20 ^**^	−0.07	−0.11	(0.87)		
6. Vitality	3.85	0.57	0.04	0.35	−0.09	−0.02	0.44 ^**^	(0.86)	
7. COB	3.90	0.56	0.09	0.16	−0.12 ^*^	0.06	0.26 ^**^	0.47 ^**^	(0.88)

*N* = 288. COB = Customer-oriented behavior. SD = standard deviation. ^*^
*p* < 0.05, ^**^
*p* < 0.01. Cronbach’s alphas in brackets on the diagonal.

**Table 3 ijerph-17-02296-t003:** Reliability and convergent validity check.

Variables	Variables Cronbach’s Alpha	rho_A	Composite Reliability (CR)	Average Variance Extracted (AVE)
Servant leadership	0.87	0.88	0.92	0.76
Vitality	0.86	0.87	0.90	0.69
COB	0.88	0.88	0.93	0.73

**Table 4 ijerph-17-02296-t004:** Regression analysis results for mediating effect (Study 1).

	Model 1 COB	Model 2 Vitality	Model 3 COB	Model 4 COB
Gender	−0.91	−0.4	−0.07	−0.7
Age	0.00	0.00	−0.00	0.00
Education	−0.07	−0.04	−0.52	0.05
Work tenure	0.00	0.00	0.01	0.01
Servant leadership	0.22 ^***^	0.39 ^***^		0.06
Vitality			0.45 ^***^	0.43 ^***^
R^2^	0.09	0.21	0.24	0.24
ΔR^2^	0.07 ^***^	0.20 ^***^	0.21 ^***^	0.15 ^***^
F	5.44 ^***^	14.32 ^***^	16.97 ^***^	14.36 ^***^
ΔF	20.09	68.22	76.33 ^***^	53.80

*N* = 288. COB = Customer-oriented behavior. ^*^
*p* < 0.05, ^**^
*p* < 0.01, *** *p* < 0.001 Regression coefficients represent unstandardized parameters.

**Table 5 ijerph-17-02296-t005:** Sample distribution (Study 2).

Variables	Value	Frequency	Percent
Employees’ gender	Male	107	58.8%
	Female	75	41.2%
Employees’ education	High school/technical school	87	47.8%
	Associates degree	65	35.7%
	Bachelors degree	29	15.9%
	Masters degree and above	1	0.5%
Employees’ work tenure	<5	138	75.8%
	5–9	28	15.4%
	10–14	8	4.4%
	>15	8	4.4%
Leaders’ gender	Male	20	64.5%
	Female	11	35.5%
Leaders’ education	Associates degree and below	8	25.8%
	Bachelors degree	18	58.1%
	Masters degree and above	5	16.1%
Leaders’ tenure in a management role	<5	23	74.2%
	5–9	7	22.7%
	>10	1	3.2%

**Table 6 ijerph-17-02296-t006:** Descriptive statistics, reliabilities, and correlations of variables (Study 2).

Individual-Level Variables	Mean	SD	1	2	3	4	5	6	7
1. Gender	1.59	0.49							
2. Age	25.17	4.14	0.19 ^*^						
3. Education	1.65	0.70	0.03	0.53 ^**^					
4. Work tenure	3.31	3.83	0.17 ^*^	0.45 ^**^	0.21 ^**^				
5. Vitality	3.98	0.66	0.24 ^**^	0.23 ^**^	0. 05	0.02	(0.87)		
6. Organizational identification	3.76	0.72	0.10	0.07	0.03	0.67	0.62 ^**^	(0.89)	
7. COB	4.01	0.65	0.16 ^*^	0.13	0. 06	0.10	0.80 ^**^	0.68 ^**^	(0.86)
Team-level variables	Mean	SD	1	2	3	4	5		
1. Leaders’ gender	1.35	0.49							
2. Leaders’ age	30.71	3.84	0.08						
3. Leaders’ education	1.90	0.65	0.11	0.24 ^**^					
4. Leaders’ work tenure	4.16	2.66	0.11	0.55	0.43 ^*^				
5. Servant leadership	3.95	0.63	0.16	0.25	0.13	0.16	(0.86)		

*N* = 182 team members (level 1), *N* = 31 teams (level 2). COB = Customer-oriented behavior. ^*^
*p* < 0.05, ^**^
*p* < 0.01. Cronbach’s alphas in brackets on the diagonal.

**Table 7 ijerph-17-02296-t007:** Reliability and convergent validity check.

Variables	Variables Cronbach’s Alpha	rho_A	Composite Reliability (CR)	Average Variance Extracted (AVE)
Servant leadership	0.86	0.87	0.91	0.77
Vitality	0.87	0.88	0.92	0.70
Organizational identification	0.86	0.86	0.92	0.71
COB	0.86	0.88	0.90	0.69

**Table 8 ijerph-17-02296-t008:** Results of HLM for main and mediation effects (Study 2).

Variables	Model 1 Vitality	Model 2 Organizational identification	Model 3 COB	Model 4 COB	Model 5 COB
*Level 1*					
Gender	0.15	0.05	0.09	0.01	0.04
Age	0.01 *	0.01	0.01	0.00	0.00
Education	0.05	0.03	0.05	0.08	0.09 *
Work tenure	0.01	0.00	0.03 *	0.02 *	0.02 *
Vitality		0.50 ***		0.58 ***	0.56 ***
Organizational identification	0.28 ***			0.23 ***	0.19 ***
*Level 2*					
Leaders’ gender	0.09	0.23	0.24	0.05	0.09
Leaders’ age	0.00	0.02	0.02	0.00	0.01
Leaders’ education	0.14	0.17	0.31 *	0.07	0.12
Leaders’ tenure	0.01	0.01	0.02	0.01	0.01
Servant leadership	0.29 *	0.29 *	0.53 ***		0.21 **

*N* = 182 team members (level 1), *N* = 31 teams (level 2). Unstandardized estimates are reported. Values in parentheses are robust standard errors.* *p* ≤ 0.05; ** *p* ≤ 0.01; *** *p* ≤ 0.001 (two-tailed test).

**Table 9 ijerph-17-02296-t009:** Conclusion of testing all the hypotheses.

Hypotheses	Results
Hypothesis 1. Servant leadership is positively related to employees’ customer-oriented behavior (Study 1 and Study 2).	Supported
Hypothesis 2. Servant leadership is positively related to employees’ vitality (Study 1 and Study 2).	Supported
Hypothesis 3. Vitality is positively related to employees’ customer-oriented behavior (Study 1 and Study 2).	Supported
Hypothesis 4. Vitality mediates the relationship between servant leadership and employees’ customer-oriented behavior (Study 1 and Study 2).	Supported
Hypothesis 5. Servant leadership is positively related to employees’ organizational identification (Study 2).	Supported
Hypothesis 6. Organizational identification is positively related to employees’ customer-oriented behavior (Study 2).	Supported
Hypothesis 7. Organizational identification mediates the relationship between servant leadership and employees’ customer-oriented behavior (Study 2).	Supported

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
