# Peer review of "How Servant Leadership Leads to Employees’ Customer-Oriented Behavior in the Service Industry? A Dual-Mechanism Model"

_ijerph, 2020, doi:10.3390/ijerph17072296_

Round 1
Reviewer 1 Report
A good piece of writing. I have gone through the whole paper and understood the significance. Some of my comments are given below:
- Clearly enlighten the motivation behind the study, properly describe the concept of servant leadership.
- Theory discussion is very limited therefore, it is necessary to separate theory and hypothesis into two sections.
- Sample selection must be described more sophisticated way following the Masud et al. 2019.
- Authors followed the CFA model there for it is necessary to show the graphical presentation of the conceptual model (see Masud et al. 2019).
- The variable explanation is not clear please explain dependent, independent, mediator, and control variables.
- Include factor loading value for reliability test.
- Include the whole questionnaire in the appendix.
- Update the paper using the most current referencing.
Masud, M.A.K; Harun, M; Khan, T; Bae, S; and Kim, J.D. (2019). Organizational Strategy and Corporate Social Responsibility: The Mediating Effect of Triple Bottom Line, International Journal of Environmental Research and Public Health, 16(22), 4559.
Author Response
RESPONSES TO THE COMMENTS OF REVIEWER 1:
(AUTHOR RESPONSES ARE ITALICIZED)
A good piece of writing. I have gone through the whole paper and understood the significance. Some of my comments are given below:
Response: Thank you very much for your positive and encouraging comments on our revised manuscript. We really appreciate that your comments and suggestions help us a lot. We have revised our paper in accordance with your comments and suggestions. Below are our point-to-point responses to your concerns.
1. Clearly enlighten the motivation behind the study, properly describe the concept of servant leadership.
Response: Thank you for this valuable comment. We make the following revisions:
(1). In the Introduction section, we explicitly added our research motivation by arguing that although researchers have suggested the potential influences of servant leadership on followers’ cognition and affect, existing studies failed to empirically address this issue; therefore, we draw on the cognitive affective processing system theory (CAPS) to unfold the processes where servant leadership leads to COB through facilitating organizational identification and vitality. Specifically, we added that “Recent reviews in servant leadership literature highlight that servant leaders can facilitate followers’ decriable outcomes at work through enhancing their positive affect and cognitions (e.g., Lu et al., 2019; Zhang et al., 2012), and further call for empirical examinations on opening the black box of why and how servant leadership exerts influences through these mechanisms due to quite few studies pinpointing this issue (Eva et al., 2019). To fill in this research gap, in this study, we tackle the complex intervening mechanism of affect and cognitions between empowering leadership and employees’ COB in the service industry.” (p. 2).
(2). In the new manuscript, we added a new section (i.e., 2. Research Background and Theoretical Discussion) (p. 3-4) to describe the concept of servant leadership empirically and theoretically. For example, we introduced servant leadership by inserting the original idea from Greenleaf, and then the development of the servant leadership theory. In addition, we provided some empirical evidence to indicate the influences of servant leadership on different outcomes, especially in the service industry.
2. Theory discussion is very limited therefore, it is necessary to separate theory and hypothesis into two sections.
Response: Thank you for this valuable comment. We followed your advice to add a new section (i.e., 2. Research Background and Theoretical Discussion) (p. 3-4) before the hypothesis section. Specially, in the new research background and theoretical discussion section, we first presented the concepts of servant leadership in general and provided more evidence to show the development of servant leadership theory (this is consistent with the first comment above). Moreover, we introduced the theoretical lens in the current research (i.e., CAPS) to argue our theoretical background.
3. Sample selection must be described more sophisticated way following the Masud et al. 2019.
Response: Thanks for the very helpful comment! In the new manuscript, we followed Masud et al.’s (2019) paper and added more details on how we collected date in both Study 1 and Study 2 (p. 7, 9). For example, in Study 1, we inserted that “We first contacted the CEOs/HR officers of these companies to obtain permission for our investigation. Next, employees received a brief introduction explaining that all information provided would be kept confidential, and the results would be sent to the researchers only. To avoid response bias, the names of the measures were not revealed, and the survey was anonymous. Informed consent was obtained from all participants to ensure that the researchers had the right to use the collected data.”.
4. Authors followed the CFA model there for it is necessary to show the graphical presentation of the conceptual model (see Masud et al. 2019).
Response: Your Thank you for this valuable comment. We in the new manuscript added two tables (i.e., Table 1 and Table 5) (p. 7, 10) to show the detailed information about our sample distribution in both studies.
5. The variable explanation is not clear please explain dependent, independent, mediator, and control variables.
Response: Your point is well-taken and we clarified the dependent, independent, mediator, and control variables in the measurement section and the results section (p. 7-8, 10).
6. Include factor loading value for reliability test.
Response: Thank you for this good comment. We added the results of factor loading in the appendix (p. 17).
7. Include the whole questionnaire in the appendix.
Response: Thank you for this valuable comment. We added the whole questionnaire of both studies in the appendix (p. 17).
8. Update the paper using the most current referencing.
Response: Thank you for this comment. We added current referenced in the new manuscript, such as Hartnell, et al., in press, Lee et al., 2020, Luu et al., 2020, and Qiu et al., (in press). These references are highly relevant to our current research.
Reviewer 2 Report
Review – 2020-03-16 – Servant Leadership and customer-oriented behavior
General Comment
This is a very interesting article, that is a proposal to capture the link between leadership style and customer orientation of employees. It is supported by a quantitative statistical analysis based on a questionnaire addressing elements of leadership style (servant leadership), customer orientation of employees, and, inspired by the CAPS, two mediations, the organizational identification and vitality. At this stage, one should stress that there are several additional theories addressing the motivation of employees, and leadership styles, e.g. employee empowerment.
The article presents enough evidences and justifications for the elaboration of the 7 hypothesis that are used throughout the text.
While the 2 studies are well explained, including their respective limitations, it is of course not possible to track the validity of the conclusions, as neither the content of the questionnaires, nor the data are available.
But the article is presenting a clear contribution on methodologies that can inspire further research in this topic.
Because there is definitively potential to further explore this topic, the conclusion could be expanded with more proposals for further researc
To 1: Introduction.
The role of servant leadership is well presented. It could still relate to other concepts that are typically used in HR (see above).
Also:
L56: SAPS or CAPS???
L 60-63: How CAPS is supporting the design of this study could be better explained
L64: vitality: is this really influenced by leadership style, or is this an intrinsic quality of employee, that bad management can only spoil (numerous HR study supporting this, by the way): elaborate.
To: 2.1: Servant leadership.
Text could elaborate on how to qualify and measure it.
L124: ‘pay more’: find a better expression
To 2.2: Vitality
Description and quantification could be improved, as this is quite central to the study.
Vitality : description as ‘physical strength’, etc…: not convincingly comprehensive.
Comment on para 2:
The links between servant leadership and COB are well described, but this is rather one facet of the story of employee motivation, and addressing their needs, such as esteem, dignity, etc (beyond the Maslow model). The text would have a better impact by at least referring to these.
To 3: Study 1
The text does not discuss the validity and limitations of these self-assessments, e.g. stemming from personality bias, etc…
To 4: study 2:
Supervisory assessment is not necessarily more objective: personality biais as well, personality identification, role model or not, etc…
General comment to 3 and 4: the article seems to rely on rather solid statistical analysis, but it is not possible for a reader to check and validate the conclusions. Also: a table with the list of hypothesis and their validity would have certainly improved the readability of the text
To 5: discussion
Convincing argument about multiple mediations
Fostering servant leadership style could be expanded (e.g.: often fostered through 360o assessment, and employees motivation surveys, with feed-back to management, and embedding results in yearly objectives.
Author Response
RESPONSES TO THE COMMENTS OF REVIEWER 2:
(AUTHOR RESPONSES ARE ITALICIZED)
This is a very interesting article, that is a proposal to capture the link between leadership style and customer orientation of employees. It is supported by a quantitative statistical analysis based on a questionnaire addressing elements of leadership style (servant leadership), customer orientation of employees, and, inspired by the CAPS, two mediations, the organizational identification and vitality. At this stage, one should stress that there are several additional theories addressing the motivation of employees, and leadership styles, e.g. employee empowerment.
Response: Thank you very much for your positive and encouraging comments on our revised manuscript. We really appreciate that your comments and suggestions help us a lot. We have revised our paper in accordance with your comments and suggestions. Below are our point-to-point responses to your concerns.
The article presents enough evidences and justifications for the elaboration of the 7 hypothesis that are used throughout the text.
While the 2 studies are well explained, including their respective limitations, it is of course not possible to track the validity of the conclusions, as neither the content of the questionnaires, nor the data are available.
Response: Thank you for this valuable comment. We added the whole questionnaire of both studies in the appendix (p. 17). In addition, we acknowledged that future research is highly encouraged to replicate our results with other data set; therefore, we add this as a limitation (p. 14).
But the article is presenting a clear contribution on methodologies that can inspire further research in this topic.
Response: Thank you for this valuable comment. We added the whole questionnaire of both studies in the appendix (p. 17).
Because there is definitively potential to further explore this topic, the conclusion could be expanded with more proposals for further research
Response: Thank you for this valuable comment. We added the whole questionnaire of both studies in the appendix (p. 17).
To 1: Introduction.
The role of servant leadership is well presented. It could still relate to other concepts that are typically used in HR (see above).
Response: Thank you for this valuable comment. We added a new section (i.e., 2. Research Background and Theoretical Discussion) (p. 3-4) to describe the concept of servant leadership empirically and theoretically. More importantly, in this section, we clearly highlighted that servant leadership is relevant to but different from other HR practices and/or leadership approaches.
Also:
L56: SAPS or CAPS???
Response: Thanks for this comment and we are sorry to make the ambiguities. In the new manuscript, we corrected this into “CAPS” (p. 1).
L 60-63: How CAPS is supporting the design of this study could be better explained
Response: Thank you for this valuable comment. First, in the introduction, we provided the research rationale by utilizing CAPS. Specifically, we argued that scholars have suggested to explore the mechanisms of (servant) leadership and outcomes by considering multiple mediators, existing research failed to address this limitation yet. Next, in the new section of theory discussion (p. 4), we claimed that CAPS is a well-established theoretical framework in leadership studies since leadership acting as a situational factors can stimulate followers’ emotional and cognitive units towards corresponding outcomes. Afterwards, we put more evidence to support the legitimacy of utilizing CAPS in our current research.
L64: vitality: is this really influenced by leadership style, or is this an intrinsic quality of employee, that bad management can only spoil (numerous HR study supporting this, by the way): elaborate.
Response: Thank you for this good comment. In the new manuscript, we added this as a limitation by not only declaring that leadership are also likely to be detrimental to employee vitality, but also providing some empirical evidence from previous studies. Specifically, we inserted that “Relatedly, regarding to the mediating effect of vitality, although the current study, consistent with previous research, indicated that leadership styles can influence subordinates’ vitality (e.g., Binyamin & Brender-Ilan, 2018), scholars have found that bad management are likely to spoil vitality. For example, Pick and coauthors (2015) evidenced that leaders and human resources department providing training and development are unlikely to foster employees’ vitality because these employees may experience training as too-much demanding or too-time consuming. Therefore, future empirically studies are highly encouraged to investigate the potential dark side of leadership on employee vitality.” (p. 14).
To: 2.1: Servant leadership.
Text could elaborate on how to qualify and measure it.
Response: Thank you for your comment. We, in the new added section of research background and theoretical discussion (p. 3-4), specifically explained the conceptualization and the operationalization of servant leadership following the theoretical line of Liden et al. (2014).
L124: ‘pay more’: find a better expression
Response: Thank you for this suggestion. We rewrote the sentence “put more efforts into completing work”(p. 4).
To 2.2: Vitality
Description and quantification could be improved, as this is quite central to the study.
Response: Thank you for this comment. In the new manuscript, we added the concpetulization of vitality by inserting that “It can be generally defined as an affective component of thriving that refers to individuals’ positive and affective feelings of energy, aliveness, and function (Kark and Carmeli, 2009).” (p. 5). Moreover, we provided more empirical and theoretical evidence to highlight the qualification of vitality in the past research. Specifically, we added that “Previous studies have indicated that when individuals have a high level of vitality, they obtain the “experience of having energy available to one’s self”(Tummers et al., 2016, p. 356), which is associated with a range of positive health and wellness indicators(e.g., work engagement and overall well-being) (Van Scheppingen et al., 2015).” (p. 5).
Vitality : description as ‘physical strength’, etc…: not convincingly comprehensive.
Response: Thank you for pointing this. We corrected this as “psychological strength”.
Comment on para 2:
The links between servant leadership and COB are well described, but this is rather one facet of the story of employee motivation, and addressing their needs, such as esteem, dignity, etc (beyond the Maslow model). The text would have a better impact by at least referring to these.
Response: Thank you for pointing this. We corrected this as “psychological strength”.
To 3: Study 1
The text does not discuss the validity and limitations of these self-assessments, e.g. stemming from personality bias, etc…
Response: Thank you for this comment. In the limitation section, we added the self-assessment as a limitation. Specifically, we added that “Moreover, most of our data, including the COB measure, are self-reported from employees, leading to the possibility of bias of self-assessment stemming from personality bias.” (p. 8).
To 4: study 2:
Supervisory assessment is not necessarily more objective: personality biais as well, personality identification, role model or not, etc…
Response: Thank you for pointing this. Consistent with the comment above, we discussed this limitation in the new manuscript by inserting that “In the current research, we relied on subjective assessment of COB (employee self-rated in Study 1 and supervisor-rated in Study 2), which failed to entirely preclude the possibility of personal bias.” (p. 14).
General comment to 3 and 4: the article seems to rely on rather solid statistical analysis, but it is not possible for a reader to check and validate the conclusions. Also: a table with the list of hypothesis and their validity would have certainly improved the readability of the text
Response: Your comment is well-taken and we added a table to make a conclusion of all the hypotheses (see Table 9) (p. 12).
To 5: discussion
Convincing argument about multiple mediations
Response: Thank you for this valuable suggestion. In the new manuscript, we added more theoretical discussions in order to convinced readers about the multiple mediations. Specifically, we first echoed to the theoretical argument in the introduction section about the research gaps of examining multiple mediators from different aspects. Then, based on our findings, we discussed our contributions to the literature of servant leadership and CAPS. For example, we inserted that “Generally, we respond to scholars’ recent calling that “by examining multiple … mediators concurrently, we can rule out some of these effects and build a more parsimonious and useful picture of what is going on.” (Hughes et al., 2018, p. 558) through utilizing CAPS to test two influential paths of servant leadership on employee COB. Specifically, as CAPS theoretically indicates that situational factors can trigger individuals’ cognitive and affective characteristics towards certain behavioral outcomes (Mischel and Shoda, 1995), leadership scholars have suggested that positive leadership styles are beneficial situational factors to boost employee desirable outcomes at work through influencing employees’ cognitions and affects (e.g., Yuan et al., 2019). The current research extends this line of literature to indicate that servant leadership is characterized as a “positive” leadership approach can contribute to employee positive behaviors (i.e., COB) in the service setting.” (p. 13).
Fostering servant leadership style could be expanded (e.g.: often fostered through 360o assessment, and employees motivation surveys, with feed-back to management, and embedding results in yearly objectives.
Response: Thank you for this comment. In the new manuscript, we added more contents of practical implications. Specifically, we inserted that “Meanwhile, a 360 degree leadership assessment can be launched to evaluate leader or manager’s servant-oriented behaviors and attitudes. In addition, such managerial activities as employees motivation surveys, with feed-back to management, and embedding results in yearly objectives can be installed in the service-centered organizations to further strengthen supervisors enacting servant leadership approach.” (p. 14).

Reviewer 3 Report
The manuscript reads very well. The logic is very clear and the development of the hypothesis was well formulated. All the concepts in this paper were well defined and presented. Both studies were well designed and conducted. I believe the second study using data from the supervisor increases greatly the internal validity. Therefore, I only have some suggestions for further improvement.
Please indicate the page number for direct citations. E.g., Page 1, line 40; Page 2, line 75
Page 2, line 56, what is SAPS. Please also include the full name.
Maybe include the standard deviation of the mean age for the first study.
Please report the Cronbach alpha for each scale.
Did you check the construct validity (convergent and divergent validity)?
Please show the full name (Common Method Bias) when CMB was first introduced.
Which software or statistical package did you use for the multi-level analysis? MLwiN?
Author Response
RESPONSES TO THE COMMENTS OF REVIEWER 3:
(AUTHOR RESPONSES ARE ITALICIZED)
The manuscript reads very well. The logic is very clear and the development of the hypothesis was well formulated. All the concepts in this paper were well defined and presented. Both studies were well designed and conducted. I believe the second study using data from the supervisor increases greatly the internal validity. Therefore, I only have some suggestions for further improvement.
Response: Thank you very much for your positive and encouraging comments on our revised manuscript. We really appreciate that your comments and suggestions help us a lot. We have revised our paper in accordance with your comments and suggestions. Below are our point-to-point responses to your concerns.
Please indicate the page number for direct citations. E.g., Page 1, line 40; Page 2, line 75
Response: Thank you for your suggestion. We have added the page numbers it in our new manuscript on p. 1.
Page 2, line 56, what is SAPS. Please also include the full name.
Response: Thanks for this comment and we are sorry to make the ambiguities. In the new manuscript, we corrected this into “CAPS” (p. 1).
Maybe include the standard deviation of the mean age for the first study.
Response: Thank you for your suggestion. In the new manuscript, we added the standard deviation of the mean age in Study 1 (p. 7).
Please report the Cronbach alpha for each scale.
Response: We follow your suggestion and reported the Cronbach alpha for each scale in both Study 1 and Study 2 (p. 7, 10).
Did you check the construct validity (convergent and divergent validity)?
Response: Your suggestion is well-taken. We provide more evidence to show the construct validity. First, we added the Cronbach alpha for each scale (this is consistent with the comment above). Second, we provided the evidence of CFA for each scale in both studies (p. 7, 10).In addition, we conducted analyses on internal reliability and convergent validity measures in both Study 1 and Study 2 (see Table 3 and Table 7) (p. 8, 11).
Please show the full name (Common Method Bias) when CMB was first introduced.
Response: Thank you for pointing this issue, and we have added the full name of CMB in the text (p. 8) .
Which software or statistical package did you use for the multi-level analysis? MLwiN?
Response: Thank you for this comment. We added that “we employed the hierarchical linear modeling (HLM) analyses to test hypotheses” (p. 10).

Round 2
Reviewer 1 Report
Thank you dear author. The points I have raised all answers are given accordingly. Now the paper looks good.